# Artificial Light at Night Drives Earlier Singing in a Neotropical Bird

**DOI:** 10.3390/ani12081015

**Published:** 2022-04-13

**Authors:** Oscar Humberto Marín Gómez

**Affiliations:** 1Programa de Biología, Grupo de Biodiversidad y Educación Ambiental, Grupo de Investigación y Asesoría en Estadística, Carrera 15 #12N, Armenia 630004, Colombia; oschumar@iztacala.unam.mx; 2Laboratorio de Ecología, Unidad de Biotecnología y Prototipos (UBIPRO), Facultad de Estudios Superiores Iztacala, Universidad Nacional Autónoma de México, Av. de los Barrios 1, Los Reyes Iztacala, Tlalnepantla 54090, Mexico

**Keywords:** ALAN, anthropogenic noise, dawn chorus, light pollution, song timing, urban ecology, urbanization

## Abstract

**Simple Summary:**

Urban birds have to cope with dominant stressors as anthropogenic noise and artificial light at night by adjusting their song traits. However, evidence of such adjustments has been studied thus far in temperate cities, rather than tropical cities. Here, I tested whether noise and light pollution influence earlier singing behavior in a tropical bird, the Saffron Finch. Birds in highly urbanized sites sang earlier at dawn and this timing difference was driven by light pollution instead of anthropogenic noise. Overall, these results suggest that light pollution could have a detrimental impact on the circadian rhythms of urban tropical birds such as daily singing routines.

**Abstract:**

Anthropogenic noise and artificial light at night (ALAN) can disrupt the morning singing routines of urban birds, however, its influence on tropical species remains poorly explored. Here, I assessed the association between light and noise pollution with the dawn chorus onset of the Saffron Finch (*Sicalis flaveola*) in a city in Colombia. I studied 32 sites comprised of different conditions of urban development based on built cover. I recorded the time of the first song of the Saffron Finch, the conspecific density and measured anthropogenic noise and ALAN using smartphone apps. The findings of this study show that Saffron Finches living in highly developed sites sang earlier at dawn than those occupying less urbanized sites. Unexpectedly, this timing difference was related to ALAN instead of anthropogenic noise, suggesting that light pollution could drive earlier dawn chorus in a tropical urban bird. Saffron Finches could take advantage of earlier singing for signaling territorial ownership among neighbors. Future studies need to assess the influence of ALAN on the dawn chorus timing of Neotropical urban birds.

## 1. Introduction

The daily singing routines of birds typically involve two peaks of high vocal activity performed by different individuals of multiple species around dawn and dusk [1,2,3]. Those choruses are a dominant component of the biophony of terrestrial soundscapes [4] and represent an intriguing feature of the avian natural history [2]. In particular, avian dawn choruses are characterized by a high singing activity of multiple individuals of many species typically occurring from twilight (i.e., the light from the sky between full night and sunrise to sunrise) and have received important attention due to their consequences for fitness [2,5]. Given the complexity of this phenomenon, multiple hypotheses have been proposed to explain why birds sing more at dawn than during daytime, including physiological (circadian rhythms on testosterone and melatonin); environmental (predation risk, acoustic transmission, inefficient foraging, body condition); and social factors (intrasexual and intersexual communication; see [2,5]).

Urban environments have become key scenarios in helping us understand how birds can deal and adapt to novel conditions generated by the environmental and ecological changes [6,7]. As such, avian communication in cities has received much attention in recent years, providing evidence of acoustic adjustments to urbanization (reviewed by [8]). In general, urban birds have to cope with dominant urban stressors as anthropogenic noise and artificial light by adjusting either temporal and spectral traits of their acoustic phenotype [8,9]. As noise levels peak at rush hours (mainly to traffic and pedestrian activity), having earlier singing routines is a good strategy to avoid acoustic masking [7,10,11,12,13,14]. The overlapping of the peak of anthropogenic noise and dawn chorus could also be strongly geographically dependent. Around the equator, anthropogenic noise and dawn chorus may overlap in time, however, in the far north, dawn chorus (ca 3:00–4:00 h in the breeding season) occurs a few hours before the peak of anthropogenic noise (7:00–9:00 h) [5,7,14]. Adjusting the structural traits, such as increasing the minimum acoustic frequencies or the amplitude of the vocalizations, are another strategy that can help to reduce acoustic masking and increase the active space for signaling [7,9,15,16].

Artificial light at night (ALAN) comprises of any source of anthropogenic illumination at night, both indoor and outdoor, which drives changes in the physiology and behavior of the urban wildlife [17,18,19,20,21,22]. Studies from Palearctic and Nearctic cities suggest that ALAN disrupts the daily rhythms in songbirds by altering the circadian rhythms of sleep and hormone production [19,22,23,24,25]. In particular, ALAN generates similar light levels to those observed during natural twilight periods (i.e., transition between day and night when the sun drops under the horizon) altering the circadian rhythms as a consequence of twilight period extension [26]. For instance, some bird species commonly found in urban environments can extend their foraging and vocal activity to night-time in areas with high levels of light pollution [12,27,28]. Additionally, diurnal birds exposed to ALAN tend to anticipate the onset of their dawn chorus [10,29,30]. However, the relationship between noise and light pollution on singing routines such as dawn chorusing remains poorly understood in Neotropical urban birds [7,14,31,32,33].

For instance, in a pioneer study in the city of Bogotá, Dorado-Correa et al. [14] assessed the influence of traffic noise and light pollution on the singing behavior of the Rufous-collared Sparrow (*Zonotrichia capensis*). ALAN was not related to the earlier singing behavior of Rufous-collared sparrows, however, birds living in noisy urban sites sang earlier as a strategy to avoid acoustic masking by traffic later in the morning [14]. However, a recent study evidenced the earlier dawn chorus starting time of the Vermilion Flycatcher (*Pyrocephalus rubinus*), in highly urbanized areas in Armenia, Colombia, although neither anthropogenic noise nor ALAN explained this pattern [33].

At the assemblage level, one study in the city of Xalapa (Mexico), suggested an association of anthropogenic noise with earlier dawn chorus onset and chorus peak in urban areas instead of ALAN [32], while in urban areas near airports of three regions from Brazil dawn chorus inset times were not globally affected by airport noise, suggesting species-specific responses to adjust song timing with shorter seasonal light variations [31].

Given the recent interest in the study of the potential influence of both anthropogenic noise and ALAN with daily singing routines of tropical birds, in the present study, I assessed the association between light and noise pollution with the dawn chorus onset of the Saffron Finch (*Sicalis flaveola*) in different urbanization conditions of an Andean city in Colombia. The Saffron Finch is a granivorous native species, very abundant in human-modified habitats, particularly in many urban settings in some South American cities, where it is very tolerant of the presence of humans [34,35], representing a clear example of urban exploiter species and is particularly well suited for studying signaling behavior due to constant vocal activity through the year.

## 2. Materials and Methods

### 2.1. Study Species

The Saffron Finch is a small granivorous bird widely spread across the lowlands (usually below 1000 m) of South America, from Colombia to Argentina [34]. This Finch inhabits pastures, semi-open areas with scattered bushes, lawns, and gardens in rural and urban areas [34,35]. It is a secondary cavity nester that also uses abandoned nests from other species, breeds throughout the year [36], and uses artificial cavities including light poles and roofs in urban areas (Marín-Gómez pers. obs.). Widely kept and popular as a cage-bird, this Finch is very tolerant to human presence and a frequent visitor in urban feeders and gardens [34,35]. Despite being a very common species, its ecology and behavior remain poorly studied [35,37,38]. Saffron Finch males often sing exposed from a conspicuous perch (branch, post, power-line) and exhibited a very large and highly variable repertoire (25 ± 1.9 syllables), with short songs (2.1 ± 0.31 s) emitted at a wide-frequency range (5.97 ± 0.12 kHz; [39]).

### 2.2. Study Sites and Sampling Design

I conducted this study in the north of Armenia, the capital of the Quindío Department, a city of 115 km^2^ with a population of 372,344 people. The city is located in Moist Forest Biome, specifically in the mountains of the central Andes of Colombia at 1350–1550 m asl., with an annual mean precipitation of 2163 mm, a mean temperature of 21.8 °C, and relative humidity ranging between 76 and 81% [40]. The duration of the day is relatively stable across the year (12 h to 12.4), and typically traffic noise occurs around sunrise, peaking at 06:30 to 07:00 h. Currently, the city is characterized by large buildings that contrast with small houses and green areas, such as parks, and corridors of remnant vegetation through the urban area [33,41,42].

I studied 32 urban sites (Appendix A) distributed in the north of the city, comprised different conditions of urban development that included sparsely areas, typically green areas and urban parks, to highly developed sites as residential and commercial areas exposed to relatively high traffic noise and light pollution levels. Sampling sites were selected from a citywide study and where Saffron Finches were previously recorded (Marín-Gómez, pers. obs.). To ensure independence, sampling sites were spaced by a minimum distance of 300 m. These sites represented different ecological conditions of the urbanization across the city, defined by the percentage of built cover within a 50 m radius buffer for each sampling site measured from Google Maps. Afterward, I classified each site based on the urban intensity as follows [43]: sparsely developed (0–33% built cover), moderately developed (34–66% built cover), and highly developed (67–100% built cover).

### 2.3. Dawn Chorus Data and Site Variables

I visited each site once from Dec 2016 to Jan 2017 to record the time when the first individual of the Saffron Finch was heard (Appendix A). Given that some songbirds can vocalize at night [12], I started the observations at nautical twilight (beginning when the sun is 12° below the horizon) of each sampling date (retrieved from the US Naval Observatory http://aa.usno.navy.mil accessed on 1 March 2020). The duration of nautical twilight was ~45 min from nautical dawn to sunrise. Then, I located the perch where each bird was first heard in order to record the maximum levels of both anthropogenic noise and artificial light at night, starting with nautical twilight. I also annotated the number of conspecific birds heard during a 5-min period. Then I measured anthropogenic noise and artificial light pollution (at the moment of where each bird was first heard) levels using two smartphone apps: Lux Meter (Crunchy ByteBox, Greifswald, Mecklenburg-Vorpommern, Germany) and Sound Meter app (Abc Apps, St. Ives, UK) for ASUS Zenfone 2 (Taipei, Beitou District, Taiwan) smartphone (the scale of measurements of the Sound Meter app varied from 10 to 120 dB, while the measurements of Lux Meter App from 0.1 to 10,000 lux). For this, I held the smartphone at 1.2 m and moved it to each cardinal point to record for a duration of one minute the maximum value of noise (dB) and then for a duration of one minute the maximum value of light pollution (lux). Although those measurements cannot reflect real values as well as those obtained by sound meter and lux meter instruments, the results obtained could be useful as a proxy to describe noise and light pollution levels of a given environment [44,45].

As noise changes with the time of day, using one single noise measurement (noise at dawn) collected in the earliest moments of the day (when anthropogenic noise levels are expectedly low compared to the levels during the busiest hours) could underestimate noise levels, particularly in highly urbanized areas where noise levels could be lower than they are, and probably not very different from noise levels at sparsely urbanized areas. To tackle this and to represent specific-site noise pollution conditions, I retrieved the available data for noise levels (day and night) from a noise map of the city [46,47].

### 2.4. Data Analyses

Noise and light pollution level variations were tested across urbanization conditions using Kruskal–Wallis tests. To assess dawn chorus onset time variation among urbanization conditions, a generalized linear model with Gaussian error distribution was fitted. Then, the Pearson correlation coefficient was used to assess collinearity among explanatory variables (ALAN, noise at dawn, at day, and at night). As noise levels at night and day were correlated (*r* = 0.79, *p* < 0.01, *n* = 32) I used the variable with a high variance for the analysis (noise levels at night). To test the association of the onset of dawn chorus with ALAN, anthropogenic noise levels and conspecific abundance, a generalized linear mixed model was fitted using a Gaussian error distribution included site as a random factor. The response variable was the dawn chorus onset time calculated as the difference between the time of the first song of the Saffron Finch relative to the time of sunrise, where negative values represent onset times before sunrise. The explanatory variables were ALAN (lux), noise levels at dawn and at night (dB(A)), and conspecific abundance. Model assumptions were validated through diagnostic plots [48,49]. All statistical analyses were performed in R.4.1.0., using the packages glmmTMB [50] and performance [49].

## 3. Results

ALAN levels were lower in sparsely urbanized areas (Kruskal-Wallis test H [2, *N* = 32] = 15.96, *p* < 0.001; Table 1; Appendix A). However, noise levels at day and night were similar across urbanization conditions (day: H [2, *N* = 32] = 1.88, *p* = 0.39; night: H [2, *N* = 32] = 2.74, *p* = 0.26; Table 1), while noise levels at dawn were higher in highly urbanized areas (H [2, *N* = 32] = 5.57, *p* = 0.06; Appendix A).

The Saffron Finch was typically a dawn singer, and chorus onset times varied from 51 min to 10 min before sunrise (−34.0 ± 12.4 min) and occurred earlier in highly urbanized sites (X^2^ = 78.8, *p* < 0.001; Figure 1).

Saffron Finches started to sing earlier in sites exposed to higher ALAN and anthropogenic noise levels (Figure 2). However, the effect of anthropogenic noise was non-significant, and only ALAN predicted the chorus onset (Table 2), suggesting that birds living in highly illuminated sites (>7.5 lux, onset: −47.2 ± 3.3 min) started to sing 20 min earlier than birds exposed to low illuminated areas (<7.5 lux, onset: −28.8 ± 10.7 min).

## 4. Discussion

Saffron Finches started their activity earlier in highly urbanized sites as previously found for other songbirds [10,12,14,18,51]. However, earlier chorus onset was not related to anthropogenic noise levels at the moment of the singing, an unexpected result given the growing body of evidence supporting anthropogenic noise as the main factor driving the timing of singing behavior in urban birds in temperate and even tropical species [7,13,14,16,32]. Discovering no association between chorus onset and noise levels does not imply that Saffron Finches are not affected by noise. Noise levels at rush hours (around sunrise and early morning) in highly urbanized areas tend to be higher than at dawn [13,16,23]. However, as the dawn chorus occurs much earlier than the peak of anthropogenic noise perhaps it is not necessary to start singing earlier. Unfortunately to test this hypothesis detailed data on noise variation from dawn to morning are needed to assess whether Saffron Finches sing earlier to avoid acoustic masking at rush hours. Additionally, the start of singing behavior is only one component of the singing routines of birds [52]. The time at which a bird broadcasts more vocalizations (chorus peak) could be also important. For instance, a recent study in a neotropical city shows no relationship between noise levels and chorus onset but chorus peak [32].

Given that the length of photoperiod tends to be more constant towards the Equator (Secondi et al., 2020), the annual reproductive cycle of most tropical birds may be less affected by variations in day length than temperate species [53,54,55,56]. Therefore, it is expected that the breeding phenology of tropical bird species could be less affected by ALAN [18]. For instance, recent studies suggest that anthropogenic noise, not ALAN, is related to the timing of singing routines in neotropical urban birds [14,32,57]. However, findings of this study show, for the first time, that high levels of ALAN were related to the earlier dawn chorus in a tropical species. Some explanations could be proposed to explain this pattern. First, some tropical birds can be sensitive to slight photoperiodic variations and use it to initiate reproductive activity [55,56], then Saffron Finches could detect variations in natural light and ALAN. Second, birds living in urban sites exposed to different levels of ALAN (0.05–6 lux) can wake up earlier and sleep less [20,24,58], meaning Saffron Finches could anticipate the onset of their daily activities by waking up earlier and then starting to sing. In fact, in the studied city Saffron Finches, as well as other common species such as the Vermilion Flycatcher (*Pyrocephalus rubinus*) and the Tropical Kingbird (*Tyrannus melancholicus*), there was evidence that showed nocturnal singing near to lamp posts (Marín-Gómez, pers. obs.).

Earlier activities driven by ALAN can affect predator–prey interactions, as well as breeding success [22,29,30]. The Saffron Finch is a very successful species in the studied city, inhabiting areas with few potential natural predators such as hawks and eagles. In fact, small owls on the genus *Glaucidium*, commonly found in other neotropical cities [57] are absent in the studied area, and therefore Saffron Finches could have little predation pressures. Studies in temperate cities indicate that an increase in apparent day length as a result of ALAN may provide more time for foraging, as well as for social signaling [20]. Earlier activities of Saffron Finches in highly illuminated areas could not be related to improving foraging times due to the fact that, as seed-eaters, this species cannot forage for insects attracted to light poles as some tropical flycatchers typically do [27]. Therefore, Saffron Finches could take advantage of earlier singing for signaling territorial ownership among neighbors, as expected by the social dynamic hypothesis [2,5]. However, results of this study should be interpreted carefully as the dawn chorus is a multifactorial phenomenon influenced by environmental factors such as cloud cover, moonlight, temperature, and food supply; as well biological factors such as the breeding season, home range size, eye-size, foraging guild, hetero-specific neighbors occurrence, and population density of conspecific neighbors [2,5,18,59,60].

Although the effect of conspecifics was statistically non-significant and negative, interactive effects among neighbors could emerge and influence earlier singing activity. For instance, Saffron Finches that began to sing early in urban areas exposed to high levels of ALAN would be heard by neighbors occupying territories in less lighted areas. Therefore, the effect of ALAN could be overestimated due to not taking into account social factors. As previous studies have tested the influence of environmental factors on dawn chorus timing in tropical birds without considering biological factors [14,31,32,33], further studies need to include the interactive effects between environmental and social factors as neighbor density to a better understanding of the impact of urbanization on daily singing routines in tropical birds.

Finally, measurements of noise and light pollution presented here could be underestimated as they were based on smartphone applications instead of precise equipment as sound and light meters. Further studies need to obtain precise measurements of ALAN and noise levels before and after dawn in order to assess the influence of urban-related factors (noise, ALAN) on daily singing routines in tropical birds.

## 5. Conclusions

In summary, findings of this study show that Saffron Finches living in highly developed sites of an Andean city sang earlier at dawn than those occupying less urbanized sites. Unexpectedly this timing difference was related to artificial lighting instead of anthropogenic noise, suggesting that artificial light could drive earlier dawn chorus in a tropical urban bird. Further studies need to take into account the influence of multiple potential drivers as meteorological (cloudiness, temperature, moon-light), ecological (body condition, food supply, predation risk), and social factors (density of neighbors) on daily singing routines in neotropical urban birds.

## Figures and Tables

**Figure 1 animals-12-01015-f001:**
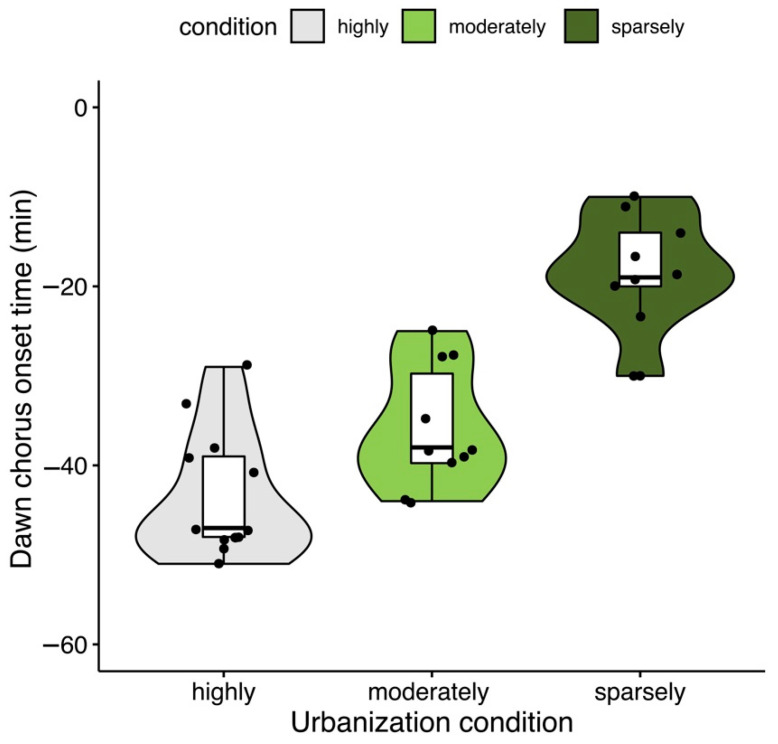
Violin plot showing the variation in dawn chorus onset times of the Saffron Finch for each urbanization condition in the city of Armenia, Colombia. Sparsely (dark green), moderately (light green), and highly developed (gray). Points indicate values of the first song by site; polygons represent the cumulative density of points.

**Figure 2 animals-12-01015-f002:**
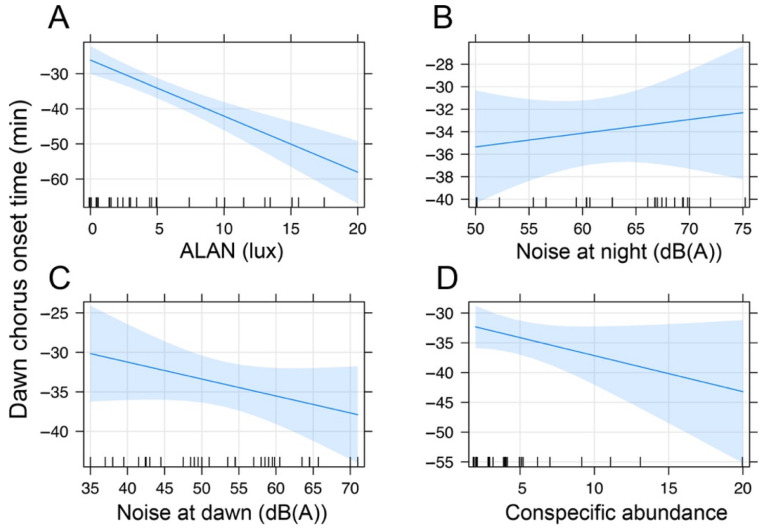
GLMM effect plot of estimated predictions between dawn chorus onset of the Saffron Finch with: (**A**) ALAN; (**B**) anthropogenic noise at night; (**C**) anthropogenic noise at dawn; and (**D**) conspecific abundance. The blue shaded areas indicate the 95% CI of the predicted values.

**Table 1 animals-12-01015-t001:** Variation in anthropogenic noise and ALAN among the studied urbanization gradient in the city of Armenia, Colombia.

Condition	Statistic	Artificial Light (lux)	Noise at Dawn (dB(A))	Noise at Day (dB(A))	Noise at Night (dB(A))
Highly (*n* = 13)	min	0.5	35.0	62.0	52.0
	max	17.5	70.0	74.0	72.0
	average ± SD	8.8 ± 5.6	55.5 ± 10.9	68.4 ± 3.7	62.9 ± 5.7
Moderately (*n* = 10)	min	0.5	44.5	62.0	50.1
	max	13.0	71.0	76.3	67.4
	average ± SD	3.5 ± 3.7	55.5 ± 7.5	66.5 ± 5.2	58.0 ± 7.5
Sparsely (*n* = 9)	min	0.0	37.0	62.0	50.1
	max	3.5	59.5	74.3	75.2
	average ± SD	0.8 ± 1.2	46.2 ± 8.3	62.0 ± 74.3	61.8 ± 10.2

**Table 2 animals-12-01015-t002:** Results of the GLMM showing the effects of ALAN, anthropogenic noise and conspecific abundance on the dawn chorus onset of *Sicalis flaveola* in the city of Armenia, Colombia. As evidenced by generalized linear mixed models, Saffron Finches showed earlier dawn chorus in sites with higher ALAN levels Neither noise levels nor conspecific abundance were related to the chorus onset (indicated by bold).

Predictors	Estimates	CI	*p*
(Intercept)	−19.31	−45.54–6.93	0.149
ALAN	−1.60	−2.13–−1.06	**<0.001**
noise at night	0.12	−0.23–0.48	0.504
noise at dawn	−0.21	−0.50–0.07	0.142
conspecifics	−0.60	−1.34–0.13	0.105

## Data Availability

The data and code presented in this study are openly available in [FigShare] at [10.6084/m9.figshare.17075150] reference number [17075150].

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
