# Peer review of "Artificial Light at Night Drives Earlier Singing in a Neotropical Bird"

_animals, 2022, doi:10.3390/ani12081015_

Round 1
Reviewer 1 Report
Dear Author,
Mountains Andes is not a biome. Biome is a major floristic unit. The author should clarify that he used the package arm for statistical analysis (as used by Dorado-Correa et al. 2016). Please describe what types of analysis (bayesian) makes this package. Because the package arm is a bayesian method, there is no model selection.Author Response
Response to Reviewer 1 Comments
1. Mountains Andes is not a biome. Biome is a major floristic unit.
Thank you very much for pointing out the deficiency. I added this suggestion as follows:
“The city is located in Moist Forest Biome, specifically in the mountains of the central Andes of Colombia at 1350-1550 m asl., with an annual mean precipitation of 2,163 mm, a mean temperature of 21.8°C, and relative humidity ranging between 76 and 81% ([40]).”
2. The author should clarify that he used the package arm for statistical analysis (as used by Dorado-Correa et al. 2016). Please describe what types of analysis (bayesian) makes this package. Because the package arm is a bayesian method, there is no model selection.
I appreciate the reviewer's suggestion. However, I did not use the package arm, because I performed generalized linear mixed models using the packages glmmTMB and performance. Specifically, I used these models to test the association of the onset of dawn chorus with ALAN, anthropogenic noise levels and conspecific abundance, including site as a random factor.

Reviewer 2 Report
The Author improved the manuscript and considered my comments and suggestions. In my opinion manuscript is interesting, well and logically written, also methodologically looks good. In my opinion manuscript should be published in Animals.
Author Response
Response to Reviewer 2 Comments
The Author improved the manuscript and considered my comments and suggestions. In my opinion manuscript is interesting, well and logically written, also methodologically looks good. In my opinion manuscript should be published in Animals.
Thank you very much for these valuable comments!

This manuscript is a resubmission of an earlier submission. The following is a list of the peer review reports and author responses from that submission.
Round 1
Reviewer 1 Report
This is very interesting paper about the effect of light and noise pollution on dawn chorus onset in tropical region. Such studies are needed to better understand birds adaptations to changing environment. The manuscript is well and logically written. However, I have a few comments which I feel should be addressed:
- Local light conditions should be described precisely; please give exact definition of dawn (civil, nautical, astronomical), time period between dawn and sunrise, time of peak of anthropogenical noise;
- Measurements of light and noise; you have used apps in your smartphone; the accuracy and precision of your measurements is unknown; I suggest some calibration of your smartphone with professional equipment;
- - Statistics: I suggest using time in a season in a model (maybe model selection?) and analysis of noise distribution across a day if such data are available.
More detailed comments:
L49-50: Please remember that overlapping of peak of anthropogenic noise and dawn chorus is strongly geographically depended. Around the equator it is true that anthropogenic noise and dawn chorus overlap in time, but in far north, dawn chorus (ca 3-4 AM in the breeding season) is a few hours before the peak of anthropogenic noise (7-9 AM). Therefore, birds do not need to start singing early in far north because noise.
L58-60: I agree but look that (1) some diurnal species sing also at night; (2) species inhabit northern regions where night is completely bright.
L67-73: I think that you should explain why different studies report various results, and which factors are responsible for these different patterns (e.g., various species can see under different level of light; predator pressure). It is important to show that various responses on pollution by light and noise are expected in different species and locations.
L80-84: Hypotheses and predictions are missing. Also, you should explain what is novel in your study?
L88: … year.
L98-99: In which part of the year the study species does breed?
L104-109: To better understand your study please describe how stable is duration of day across the year and how twilight is related to the peak of anthropological noise. Your study site is close to the equator, therefore you may expect different effect of noise and light pollution on birds behavior than in cities located far north or south.
L123-124: Is it a breeding or non-breeding season? The time in a season was not included in the analyses. This factor can be important. I cannot see (by the way, please consider including dataset as a supplementary materials) how surveys were distributed in the time and whether were randomly distributed across areas with various level of urbanization.
L 125: Is it your definition of dawn (nautical dawn, beginning when sun is 12° below the horizon)? How long is twilight in your study location (from nautical dawn to sunrise)? This information are important to understand results of your study.
L129-131: This is not professional equipment. Please give some details about accuracy and precision of measurement taken by using your smartphone and apps. Did you calibrate your device with any professional decibel or lux meters? Which settings did you use? What was the range of measurements, especially in the case of light pollution measurements?
L141: How long displacement between dawn chorus (or sunrise) and peak of anthropogenic noise is.
L145-146: In this way you were able to compare an average noise pollution between sites, not the distribution of noise across a day. I think that daily distribution of noise pollution is important in your study. Please consider mixed models or generalized estimating equations to test it. Please say precisely when you use your own measurements and when data from noise map.
L148-149: I suggest using variance inflation factor instead Pearson correlation to test multicollinearity. Even strongly correlated predictors may explain different part of variance.
L156-157: What about the time in a season?
L157: Did you consider model selection by using AIC/BIC?
L164: Please explain what do you mean by “dawn”? Is it, civil, nautical, or astronomical dawn?
L168: sunrise?
L191: Maybe when dawn chorus occurs much earlier than peak of anthropogenic noise it is not necessary to start singing earlier? Showing distribution of noise pollution across a day would help better understand your study.
L219-220: Please consider predator-prey interaction but from perspective of Saffron Finch as a prey.
Author Response
REVIEWER 1
This is very interesting paper about the effect of light and noise pollution on dawn chorus onset in tropical region. Such studies are needed to better understand birds adaptations to changing environment. The manuscript is well and logically writteHowever, I have a few comments which I feel should be addressed:
- Local light conditions should be described precisely; please give exact definition of dawn (civil, nautical, astronomical), time period between dawn and sunrise, time of peak of anthropogenical noise;
- Measurements of light and noise; you have used apps in your smartphone; the accuracy and precision of your measurements is unknown; I suggest some calibration of your smartphone with professional equipment;
- - Statistics: I suggest using time in a season in a model (maybe model selection?) and analysis of noise distribution across a day if such data are available.
Thank you very much for these valuable comments! The calibration procedure is not possible to address. After all, I have not calibrated the smartphone apps with a piece of professional equipment before obtaining data, because I had no access to this equipment. Finally, I did not use model selection because I followed the same methodology and statistical analysis suggested in a previous study. I have addressed all the issues raised one by one, as shown below. And the line number is the “Simple Markup” in the revision mode of Word.
More detailed comments:
L49-50: Please remember that overlapping of peak of anthropogenic noise and dawn chorus is strongly geographically depended. Around the equator it is true that anthropogenic noise and dawn chorus overlap in time, but in far north, dawn chorus (ca 3-4 AM in the breeding season) is a few hours before the peak of anthropogenic noise (7-9 AM). Therefore, birds do not need to start singing early in far north because noise.
I appreciate the reviewer's suggestion and added it to the text as follows:
"The overlapping of the peak of anthropogenic noise and dawn chorus could be also strongly geographically dependent. Around the equator, anthropogenic noise and dawn chorus may overlap in time, but in the far north, dawn chorus (ca 3:00-4:00 h in the breeding season) occurs a few hours before the peak of anthropogenic noise (7:00-9:00 h) [5, 7,14]."
L58-60: I agree but look that (1) some diurnal species sing also at night; (2) species inhabit northern regions where night is completely bright.
I understand the reviewer's point. However, in this manuscript, I report the general patterns of avian dawn chorus as suggested in the literature, but not the causes and consequences of night singing in birds.
L67-73: I think that you should explain why different studies report various results, and which factors are responsible for these different patterns (e.g., various species can see under different level of light; predator pressure). It is important to show that various responses on pollution by light and noise are expected in different species and locations.
This is a really nice suggestion. However, here in this manuscript, I report the patterns, but not the mechanisms or factors that explained differences in dawn chorus in urban tropical birds, because there are few studies documenting associations between dawn chorus timing with differences in noise and light pollution. In fact, the reviewers' suggestions provide very interesting insights to explore in future studies, as previously showed in a recent review (Marín‐Gómez, O. H., & MacGregor‐Fors, I. 2021. A global synthesis of the impacts of urbanization on bird dawn choruses. Ibis, 163(4), 1133-1154)
L80-84: Hypotheses and predictions are missing. Also, you should explain what is novel in your study?
I appreciate the reviewer's suggestion, but in this study, I did not follow the hypothetical deductive method, because my focus was to describe a pattern, specifically was to assess the association between light and noise pollution with dawn chorus onset of the Saffron Finch (Sicalis flaveola) in different urbanization.
L88: … year.
L98-99: In which part of the year the study species does breed?
The species breed through the year
L104-109: To better understand your study please describe how stable is duration of day across the year and how twilight is related to the peak of anthropological noise. Your study site is close to the equator, therefore you may expect different effect of noise and light pollution on birds behavior than in cities located far north or south.
Thank you very much for pointing out the deficiency. I added this suggestion as follows:
“The duration of the day is relatively stable across the year (12h to 12.4), and typically traffic noise occurs around sunrise, peaking at 06:30 to 07:00 h.”
L123-124: Is it a breeding or non-breeding season? The time in a season was not included in the analyses. This factor can be important. I cannot see (by the way, please consider including dataset as a supplementary materials) how surveys were distributed in the time and whether were randomly distributed across areas with various level of urbanization.
Thanks for this comment. Unfortunately, I have no data about the reproductive biology of the saffron finch in the studied city, in fact, this species is actively singer throughout the year and probably breed through the year, but I do not know the peak of the breeding season. Please note that I have added some details in the previous version of the manuscript about sampling locations (see Figure S1), sampling dates and data availability (see Data Availability Statement). Because I collected data during one single month, testing differences on time in a “season” is not possible due to the obtained dataset.
L 125: Is it your definition of dawn (nautical dawn, beginning when sun is 12° below the horizon)? How long is twilight in your study location (from nautical dawn to sunrise)? This information are important to understand results of your study.
Thanks for this comment. The duration of nautical twilight was 45 min approximately.
L129-131: This is not professional equipment. Please give some details about accuracy and precision of measurement taken by using your smartphone and apps. Did you calibrate your device with any professional decibel or lux meters? Which settings did you use? What was the range of measurements, especially in the case of light pollution measurements?
Thank you very much for pointing out the deficiency. Unfortunately, I have not calibrated the smartphone apps with a piece of professional equipment, because I had no access to this equipment.
L141: How long displacement between dawn chorus (or sunrise) and peak of anthropogenic noise is.
Typically, traffic noise occurs around sunrise (~6:00 h) but peaking between 06:30 to 07:00 h.
L145-146: In this way you were able to compare an average noise pollution between sites, not the distribution of noise across a day. I think that daily distribution of noise pollution is important in your study. Please consider mixed models or generalized estimating equations to test it. Please say precisely when you use your own measurements and when data from noise map.
Thanks for raising this point. You right. I compared the average noise pollution between sites, but not within sites. Unfortunately, I do not have data on the daily distribution of noise pollution by each site, so GLMM cannot be used here. As indicated in the final paragraph of the methods section I retrieved available data of noise levels (day and night) from a noise map of the city.
L148-149: I suggest using variance inflation factor instead Pearson correlation to test multicollinearity. Even strongly correlated predictors may explain different part of variance.
Thanks for this suggestion. I tried using VIF to assess multicollinearity (see figure below). However, as VIF output showed a similar trend found using Pearson coefficient, I decided to use the variable with high variance for the analysis (noise levels at night). Using both noise measurements, are redundant. I believe VIF could be very useful if I will have data of noise measurements across day and night.
L156-157: What about the time in a season?
Because I collected data during one single month, testing differences on time in a “season” is not possible due to the obtained dataset.
L157: Did you consider model selection by using AIC/BIC?
I not used model selection because I followed the same methodology and statistical analysis suggested in a previous study (Dorado-Correa, A. M., Rodríguez-Rocha, M., & Brumm, H. (2016). Anthropogenic noise, but not artificial light levels predicts song behaviour in an equatorial bird. Royal Society Open Science, 3(7), 160231)
L164: Please explain what do you mean by “dawn”? Is it, civil, nautical, or astronomical dawn?
Thank you very much for pointing out the deficiency. I explain both definitions of dawn and twilight at the beginning of the introduction. Now, this section reads as follows:
“In particular, avian dawn choruses are characterized by a high singing activity of multiple individuals of many species typically occurring from twilight (i.e., the light from the sky between full night and sunrise to sunrise) and have received important attention due to their consequences for fitness [2,5].”
L168: sunrise?
Thank you very much for pointing out the deficiency.
L191: Maybe when dawn chorus occurs much earlier than peak of anthropogenic noise it is not necessary to start singing earlier? Showing distribution of noise pollution across a day would help better understand your study.
Thanks for raising this point. I added the reviewer's suggestions to the discussion section. The section reads as follows:
“However, as the dawn chorus occurs much earlier than the peak of anthropogenic noise perhaps it is not necessary to start singing earlier. Unfortunately to test this hypothesis detailed data on noise variation from dawn to morning are needed to assess whether Saffron finches sing earlier to avoid acoustic masking at rush hours.”
L219-220: Please consider predator-prey interaction but from perspective of Saffron Finch as a prey.
Thanks for raising this point. I added the reviewer's suggestions to the discussion section. The section reads as follows:
“The Saffron Finch is a very successful species in the studied city, inhabiting in areas with few potential natural predators such as hawks and eagles. In fact, small owls on the genus Glaucidium, commonly found in other neotropical cities [57] are absent in the studied area, therefore Saffron finches could have little predation pressures”.
Reviewer 2 Report
Dear Dr Marín Gómez, this is a very interesting manuscript about a scarcely analyzed topic in the Neotropics. I have a few comments.
Environmental data collection needs more clarifications. I did not understand well when data was obtained. In addition, statistical analysis needs to be improved, but I think this not will change the results obtained.
See more comments in the attached pdf.
Best regards

Author Response
REVIEWER 2
Dear Dr Marín Gómez, this is a very interesting manuscript about a scarcely analyzed topic in the Neotropics. I have a few comments.
Environmental data collection needs more clarifications. I did not understand well when data was obtained. In addition, statistical analysis needs to be improved, but I think this not will change the results obtained.
Thank you very much for these valuable comments! I addressed the issues related to environmental data. Finally, I did not use model selection because I followed the same methodology and statistical analysis suggested in a previous study. I have addressed all the issues raised one by one, as shown below. And the line number is the “Simple Markup” in the revision mode of Word.
Specific comments:
L88: please add a point. Moreover, I would like to see hypothesis or predicitions
I appreciate the reviewer's suggestion, but in this study, I did not follow the hypothetical deductive method, because my focus was to describe a pattern, specifically was to assess the association between light and noise pollution with dawn chorus onset of the Saffron Finch (Sicalis flaveola) in different urbanization.
L107: Which biome surrounds the city?
The “bioma” is mountains Andes. Now this section reads as follows:
“The city is located in the mountains of the central Andes of Colombia”
L111: I understand the urban parks are a type of urban green area.
Thank you very much for pointing out the deficiency. Now this section reads as follows:
“Currently, the city is characterized by large buildings that contrast with small houses and green areas, such as parks, and corridors of remnant vegetation through the urban area [33,41,42].”
L122: Is this the breeding season?
As showed in the line 102 the species could breed throughout the year [36], Unfortunately, I have no data about the reproductive biology of the saffron finch in the studied city, in fact, this species is actively singer throughout the year and probably breed through the year, but I do not know the peak of the breeding season
L128: the 5-min period started with nautical twilight?
Yes, you right. Thank you. I added some detail sin the text as follows:
“Then, I located the perch where each bird was first heard in order to record the maximum levels of both anthropogenic noise and artificial light at night, started with nautical twilight.”
L129: It is not clear to me when you measured light and noise pollution. Was it when you recorded a ficnh singing or when you started at at nautical twilight?
Thank you very much for pointing out the deficiency. I measured anthropogenic noise and artificial light pollution (at the moment of where each bird was first heard).
L150. I think that you should choose the variable that it is more correlated with the response variable.
Thanks for raising this point. I considered that using both noise measurements is redundant, so I retained noise levels at night in the analysis.
L170: how did you ibtain this x2 value? Was it from comparing the model with a null model?
This result is the output of the ANOVA of the glm model:
Anova(GLM.onset,type="III")
Analysis of Deviance Table (Type III tests)
Response: start
LR Chisq Df Pr(>Chisq)
condition 78.784 2 < 2.2e-16 ***
L177. I do not see a model selection approach. Ideally, the best model should be obtained, generally by AIC comparisons or backward selection by eliminating sequentially non-signiifcant variables.
Thanks for raising this point. However, I not used model selection because I followed the same methodology and statistical analysis suggested in a previous study (Dorado-Correa, A. M., Rodríguez-Rocha, M., & Brumm, H. (2016). Anthropogenic noise, but not artificial light levels predicts song behaviour in an equatorial bird. Royal Society Open Science, 3(7), 160231)